# Biotransformation of Methoxyflavones by Selected Entomopathogenic Filamentous Fungi

**DOI:** 10.3390/ijms21176121

**Published:** 2020-08-25

**Authors:** Mateusz Łużny, Tomasz Tronina, Ewa Kozłowska, Monika Dymarska, Jarosław Popłoński, Jacek Łyczko, Edyta Kostrzewa-Susłow, Tomasz Janeczko

**Affiliations:** Department of Chemistry, Wrocław University of Environmental and Life Sciences, Norwida 25, 50-375 Wrocław, Poland; tomasz.tronina@upwr.edu.pl (T.T.); e.a.kozlowska@gmail.com (E.K.); monika.dymarska@gmail.com (M.D.); jaroslaw.poplonski@upwr.edu.pl (J.P.); jacek.lyczko@upwr.edu.pl (J.Ł.); ekostrzew@gmail.com (E.K.-S.)

**Keywords:** biotransformation, entomopathogenic fungi, methoxyflavones, 4-*O*-methyloglycosylation

## Abstract

The synthesis and biotransformation of five flavones containing methoxy substituents in the B ring: 2′-, 3′-, 4′-methoxyflavones, 2′,5′-dimethoxyflavone and 3′,4′,5′-trimethoxyflavone are described. Strains of entomopathogenic filamentous fungi were used as biocatalysts. Five strains of the species *Beauveria bassiana* (KCh J1.5, J2.1, J3.2, J1, BBT), two of the species *Beauveria caledonica* (KCh J3.3, J3.4), one of *Isaria fumosorosea* (KCh J2) and one of *Isaria farinosa* (KCh KW 1.1) were investigated. Both the number and the place of attachment of the methoxy groups in the flavonoid structure influenced the biotransformation rate and the amount of nascent products. Based on the structures of products and semi-products, it can be concluded that their formation is the result of a cascading process. As a result of enzymes produced in the cells of the tested strains, the test compounds undergo progressive demethylation and/or hydroxylation and 4-*O*-methylglucosylation. Thirteen novel flavonoid 4-*O*-methylglucosides and five hydroxy flavones were isolated and identified.

## 1. Introduction

Flavonoids are common in the world of plants as their secondary metabolites, mainly in citrus fruits, olive oil, seeds [1] and vegetables [2]. They function as pigments in coloration of plants [3,4], root growth regulators, and plant defense reaction agents, and are involved in the transport of auxins between plant cells [5]. Flavonoids are present in our diet and have desired pro-health activities, including antioxidant, anti-inflammatory, anti-cancer [3,6,7], antibacterial, antifungal and antiviral activities [6,8]. Some methoxyflavones (e.g., 3′-methoxyflavone) have the ability to prevent parthanatos [9]. It is estimated that we consume a maximum of about 1 g of flavonoids a day, depending on the type and amount of meals consumed [10].

Methoxyflavones have well-described anti-cancer properties and are often more active than flavones without any functional groups [11]. Due to the presence of methoxyl groups, these compounds show stronger lipophilic properties than flavonoids with hydroxyl groups, which directly affects their bioavailability [12]. The best natural source of methoxyflavones described so far is cowslip (*Primula veris* L.), which contains these compounds in the rhizomes, roots, flowers and leaves, and from which mainly expectorants are prepared [13]. The therapeutic potential of flavonoids is often limited by their low solubility and bioavailability.

Aglycons are characterized by very low solubility in water [14]; however, addition of a polar moiety such as sugars may increase their activity, bioavailability [15] and stability [10]. For example, the water solubility of α-glucosyl-isoquercitrin increases more than 80,000-fold compared to its parent compound, quercetin [16,17]. Isoquercitrin glycoside shows 18 times higher bioavailability than quercetin after oral administration to rats [18], and much higher bioavailability compared to quercetin and isoquercitrin in human studies [19]. These results indicate that glycosyl flavones exhibit better health-promoting effects than their aglycones due to their higher bioavailability [20].

Flavonoids are not the only group of compounds in which the positive effect of the presence of a sugar moiety on the activity of the compound has been proven. The best examples are antibiotics, such as erythromycin or vancomycin, in which the presence of a sugar group is crucial for their high activity [10]. The presence and number of glycosyl substituents in the flavonoid molecule, depending on their position, should exhibit different, although not necessarily positive, properties. Naturally glycosylated derivatives of flavonoids occur widely in the world of plants, but their concentration in cells very often is relatively low [21], which makes their extraction difficult. New methods of obtaining glycosylated compounds on a larger scale are still necessary.

Carrying out the biotransformation of flavonoids to their glycosylated derivatives is less complicated and more effective than their *de novo* chemical synthesis [14], which makes biotransformation a better and cheaper alternative.

The biotransformation of five methoxyflavones with substituents on the B ring is described: 2′-(**1**), 3′-(**2**), 4′-(**3**)-methoxyflavones, 2′,5′-dimethoxyflavone(**4**) and 3′,4′,5′-trimethoxyflavone(**5**). Listed compounds were biotransformed by the strains of entomopathogenic filamentous fungi belonging to the species *Beauveria bassiana* KCh J1.5, J2.1, J3.2, J1, BBT, *Beauveria caledonica* KCh J3.3, J3.4 [21], *Isaria fumosorosea* KCh J2 [22] and *Isaria farinosa* KCh KW 1.1 [21]. Fungi of these species are among the most used for biotransformation, along with *Aspergillus niger*.

Previous studies have shown the capacity for unique 4″-*O*-methylglycosylation of hydroxyflavones observed in entomopathogenic filamentous fungal culture [15,23]. The strain *Beauveria bassiana* AM 278 catalyzed the methylglucose attachment to the hydroxyl group of a flavonoid molecule at the C-7 and C-3′ positions. Similarly, the application of *Absidia coerulea* as the biocatalyst resulted in the formation of glucosides with a glucose present at the C-7 and C-3′ positions of the same flavonoid compounds [24,25,26]. An analogous preference to the selective 4-*O*-methylglycosylation of the hydroxyl group located at the C-7 carbon was observed during the biotransformation of unique prenylated flavonoids obtained from the hop plant (*Humulus lupulus*) in cultures of *Beauveria bassiana* AM 446 and AM 278 [27,28,29,30]. *Beauveria bassiana* ATCC 13144 is capable of simultaneous 4-*O*-methylglycosylation of the hydroxyl group located at the C-3 and hydroxylation of the C-4′ carbon [31]. Furthermore, other entomopathogenic strains are capable of simultaneous hydroxylation and glycosylation of flavonoids. This functionalization takes place mainly at the C-4-OH position of the sugar moiety. Effective 4-*O*-methylglycosylation of 3-hydroxyflavone, 6-hydroxyflavone, 7-hydroxyflavone, baicalein, quercetin, naringenin, luteolin, diosmetin, and daidzein in entomopathogenic filamentous fungal culture of the genus *Isaria* has been reported [10,14,22]. Moreover, the formation of appropriate glycosides in the cultures of *Isaria fumosorosea* KCh J2, *I. farinosa* KCh J1.4, *I. farinosa* KCh J1.6 and *I. farinosa* KCh KW1.2 cultures was preceded by hydroxylation of the C-4′ carbon. Subsequent C-4′ hydroxylation and 4-*O*-methylglycosylation was observed during the incubation of flavone, 3-methoxyflavone, 5-hydroxyflavone, 6-methoxyflavone, 6-methoxyflavanone and 6-methylflavone in the cultures of the genus *Isaria* [14,22,32]. The capacity of the *I. fumosorosea* KCh J2 strain for simultaneous *O*-demethylation and glycosylation of 2′-, 3′- and 4′-methoxyflavanone was also described [32]. Based on this phenomenon, in this study, we biotransformed flavones with methoxyl groups (in different positions) located in the B ring obtained by chemical synthesis. Entomopathogenic fungal strains with confirmed ability of simultaneous hydroxylation/demethylation and glycosylation of flavonoid compounds were used as biocatalysts. The aim of this research was to investigate the microbial modification of the flavones with methoxyl-blocked carbons preferred for subsequent hydroxylation and glycosylation by the studied strains and to compare it to transformations of substrates without such an obstacle.

## 2. Results and Discussion

Based on the previously described results concerning the biotransformation of flavonoids in the cultures of entomopathogenic strains, we decided to use five *Beauveria bassiana* strains (KCh J1.5, J2.1, J3.2, J1, BBT), two *Beauveria caledonica* strains (KCh J3.3 and J3.4), and two *Isaria* strains (*I. fumosorosea* KCh J2 and *I. farinosa* KCh KW 1.1) as biocatalysts. *B*. *bassiana* species are the most common among the entomopathogenic filamentous fungi used in biotransformations, including in the case of flavonoid compounds [25,26,27]. In our previous studies concerning the biotransformation of steroids we have observed significant differences in the modification of dehydroepiandrosterone (DHEA) in the cultures of five different strains of this species [21]. Therefore, in this study, we decided to determine whether flavonoid compounds will also undergo diverse changes in the cultures of different strains of *B. bassiana*. For the first time in flavonoid biotransformations, we also tested two strains of *Beauveria caledonica* species. The *Isaria fumosorosea* KCh J2 strain has already been described in our previous study as an effective biocatalyst of flavanones and flavones as substrates [14,22,32,33]. Comparing those previously described results for *I. fumosorosea* KCh J2 to the results obtained from the transformations by the *B. bassiana* strains, we noted the similarity in the biotransformations of flavonoid compounds. In both cases, the main products are 4′-*O*-β-D-(4″-*O*-methyl)-glucopyranosides. We have also observed this ability for strains of *I. farinosa* [14,33]. In this study, we used the *I. farinosa* KCh KW 1.1 strain as a biocatalyst as well.

All used substrates, containing in their structure from one to three methoxyl groups in the B ring, were obtained by a two-step chemical synthesis. In the first stage, five chalcones were synthesized from 2′-hydroxyacetophenone and the appropriate methoxybenzaldehyde in a basic medium. Then, they were transformed into the appropriate methoxyflavones, by reaction with J_2_ in DMSO. As a result of these reactions 2′-, 3′-, 4′-methoxyflavone, 2′,5′-dimethoxyflavone and 3′,4′,5′-trimethoxyflavone were obtained.

### 2.1. 2′-Methoxyflavone (**1**) Biotransformation

Biotransformation of 2′-methoxyflavone (**1**) in the cultures of most of the tested strains resulted in one major product (**6**), with a retention time of 11.4 min according to HPLC (Table 1). Furthermore, TLC analysis confirmed that most of the biocatalysts converted 2′-methoxyflavone(**1)** to the same product. This product was identified as 2′-*O*-β-d-(4″-*O*-methylglucopyranosyl)-flavone (**6**), obtained in all tested strains except *Beauveria bassiana* KCh J1. This compound was formed as a result of subsequent demethylation and 4-*O*-methylglycosylation. However, only in the culture of the *B. bassiana* KCh J1 strain, 2′-hydroxyflavone, was an intermediate product in the biosynthesis of 2′-*O*-β-D-(4″-*O*-methylglucopyranosyl)-flavone observed. This may be evidence that the substrate demethylation process is crucial for the rate of 2′-*O*-β-D-(4″-*O*-methylglucopyranosyl)-flavone formation. Among the strains used, only the *B. bassiana* KCh J1 strain does not have the ability of 4-*O*-methylglycosylation; thus only the 2′-hydroxy flavone (**10**) was formed during the biotransformation. The highest concentrations of 2′-*O*-β-D-(4″-*O*-methylglucopyranosyl)-flavone were identified in the cultures of *Beauveria bassiana* strains: KCh J3.2 (70%); KCh J2.1 (76%); KCh BBT (87%) after 10 days of biotransformation.

The exceptions during the biotransformation of 2′-methoxyflavone (**1**) turned out to be the *I. fumosorosea* KCh J2 strain, where two additional products were also observed (it was not possible to separate compounds **7** and **8** with the selected HPLC program, so in this case UHPLC was performed) (Table 2) and the *B. bassiana* KCh J1 strain, where only one product (**10**) with a retention time of 14.8 min was observed. For this reason, a scale-up biotransformation of these two strains was performed to isolate and characterize the structure of the resulting products. As a result of the scale-up biotransformation of the 2′-methoxyflavone in the culture of the *I. fumosorosea* KCh J2, a total of four products were isolated (Figure 1). Based on NMR analyses, 2′-*O*-β-D-(4″-*O*-methylglucopyranosyl)-flavone is the major product (**6**). The formation of this compound is possible by successive demethylation and 4-*O*-methylglycosylation. The ^1^H NMR spectrum shows signals confirming that the structure of the flavone skeleton has not been changed. The presence, chemical shifts and multiplicities of the signals indicate that the only substituent is on the C-2′ carbon (as in a substrate). The structure of the flavone is confirmed by the COSY, HMQC and HMBC correlation spectra. However, instead of the signal from the protons of the CH_3_ group (visible in the ^1^H NMR substrate spectrum), signals from the sugar unit are visible. The multiplicities and positions of these signals in both the ^1^H and ^13^C NMR spectra indicate that a glucose molecule was introduced in place of the CH_3_ group. The HMBC spectrum shows the coupling of the signal from the H-1″ sugar unit proton with the signal from the C-2′ carbon of the flavone skeleton. This coupling indicates the exact place of attachment of the sugar substituent. Additionally, the ^1^H NMR spectrum shows a singlet derived from three protons (3.46 ppm). In the HMBC spectrum, this signal is coupled to the C-4″ carbon (sugar substituent), which indicates that the substituent is 4″-*O*-methylglucopyranoside.

In the ^1^H NMR spectrum of the next product (**8**) (isolated with a yield of 20%), obtained by scale-up biotransformation of 2′-methoxyflavone in the culture of *I. fumosorosea* KCh J2, there are signals indicating that the product is 4-*O*-methylglycoside. However, the presence of a singlet derived from three protons at 3.90 ppm and its coupling to the C-2′ carbon of the flavone skeleton, visible in the HMBC spectrum, prove that the methoxy group has been preserved. Based on the position and multiplicities of the signals visible in the ^1^H NMR spectrum and the couplings visible in the correlation spectra, it can be concluded that an additional substituent was attached at the C-5′ carbon. Additionally, in the HMBC spectrum, the signal from this carbon is coupled with a doublet from the H-1″ proton of the sugar unit. Based on these data, this compound was identified as 5′-*O*-β-D-(4″-*O*-methylglucopyranosyl)-2′-methoxyflavone (**8**).

Based on NMR data, another product was also identified as a glycoside derivative of 2′-methoxyflavone. Based on the chemical shifts of the signals visible in the ^1^H NMR spectrum of this compound and the couplings visible in the correlation spectra, it can be concluded that the substituent is 4″-O-methylglycoside, bound to carbon C-8. This compound was isolated in a yield of 4% and identified as 8-*O*-β-D-(4″-*O*-methylglucopyranosyl)-2′-methoxyflavone (**7**). Another product, identified as 3-*O*-β-D-(4″-*O*-methylglucopyranosyl)-2′-methoxyflavone (**9**), was isolated, but with a very low yield (<1%). From the ^1^H NMR spectrum it was found that the structure of 2′-methoxyflavone was retained. At the same time, as in the case of previously described products, signals from 4″-O-methylglycoside are visible. The lack of a signal of the H-3 proton indicates that the sugar substituent was introduced at the C-3 carbon by *O*-glycosylation reaction. Due to the low concentration of this compound, it was not possible to perform ^13^C NMR analysis and correlation spectra confirming its structure. Compounds **6**, **7, 8** and **9** are the result of successive hydroxylation and 4-*O*-methylglycosylation.

As a result of up-scale biotransformation of the 2′-methoxyflavone in the culture of the *Beauveria bassiana* KCh J1 strain, one product was isolated with 43.5% yield. On the basis of NMR analyses it was identified as 2′-hydroxyflavone (**10**) (Figure 2). This compound was formed as a result of the *O*-demethylation of 2′-methoxyflavone. The ^1^H NMR spectrum shows signals confirming that the structure of the flavone skeleton has remained unchanged. The presence and chemical shifts of the signals indicate that the hydroxyl group is on the C-2′ carbon. The structure of the flavone is confirmed by the COSY, HMQC and HMBC correlation spectra. However, instead of the signal from the three protons of the CH_3_ group (visible in the ^1^H NMR substrate spectrum) at 10.03 ppm, the signal from the proton of the hydroxyl group located at C-2′ is visible.

### 2.2. 3′-Methoxyflavone (**2**) Biotransformation

The next substrate, 3′-methoxyflavone (**2**), was converted much faster by all tested biocatalysts. It is most likely related to easier access of the enzymes responsible for the demethylation process to the methoxy group situated meta in relation to the chromene substituent (1-benzopyran or chromene). Conversion of 3′-methoxyflavone in the cultures of four out of nine biocatalysts (*I. fumosorosea* KCh J2, *I. farinosa* KCh KW1.1, *B. bassiana* KCh J1.5 and KCh J3.2) was close to 100% after three days of incubation (Table 3).

Similar to the biotransformation of 2′-methoxyflavone (**1**), during the transformation of 3′-methoxyflavone (**2**) in the culture of the *B. bassiana* KCh J1 strain, one of the main products was an effect of demethylation and was identified as 3′-hydroxyflavone (**11**) (Figure 3). All NMR data (Appendix A) confirm the structure of this product. This compound was also identified in trace amounts in the culture of the *B. bassiana* KCh BBT strain.

The major product of the transformation of 3′-methoxyflavone in most of the studied strains was the product of progressive demethylation and 4-*O*-methylglycosylation, that is 3′-*O*-β-D-(4″-*O*-methylglucopyranosyl)-flavone (**12**) (Figure 4). The chemical shifts and multiplicities of the signals visible in the ^1^H NMR spectrum confirm that the structure of the flavone skeleton has been preserved; the only substituent is on the C-3′ carbon (as in the substrate). The structure of the flavone is also confirmed by the ^13^C NMR spectra and the correlation with COSY, HMQC and HMBC spectra. The HMBC spectrum shows the coupling of the signal from the H-1″ proton of the sugar unit with the signal from the C-3′ carbon of the flavone skeleton. The ^1^H NMR spectrum shows a singlet from the three protons (3.47 ppm). In the HMBC spectrum, this signal is coupled to the C-4″ carbon (sugar substituent), which proves that the substituent is 4″-O-methylglucopyranoside.

### 2.3. 4′-methoxyflavone (**3**) Biotransformation

High conversions of 4′-methoxyflavone (**3**) in the cultures of the tested strains were comparable to those obtained for 3′-methoxyflavone (**2**) at the seven days of biotransformation. Nearly 100% conversion was observed in the culture of *Isaria farinosa* KCh KW 1.1 and four strains of *Beauveria bassiana* (KCh J1.5, J2.1, J3.2, BBT) (Table 4.). Similar to the results of the two abovedescribed compounds (**1** and **2**), the exception is the *Beauveria bassiana* KCh J1 strain, from the culture of which only the demethylation product 4′-hydroxyflavone was isolated (**13**) (Figure 5). Spectral data for this product are provided in Appendix A. In the culture of the *Beauveria bassiana* KCh J1 strain, the formation of several other products was observed after the seventh day of transformation, but they were formed concentrations that were too low to determine their structure.

As a result of scale-up biotransformation with 100 mg of 4′-methoxyflavone (**3**) in the culture of the *Beauveria bassiana* KCh J1.5 strain, 36.5 mg of 4′-O-β-D- (4″-O-methylglucopyranosyl)-flavone (14) was isolated (Figure 6). Its structure was confirmed based on spectroscopic data (Appendix A).

Comparing the biotransformation rate of substrates having a single-methoxy group, it is seen that 3′-methoxyflavone undergoes the fastest demethylation. In the literature to date, in the processes of hydroxylation and 4″-*O*-methylglucosylation observed in cultures of entomopathogenic strains, in most cases, functionalization took place at the C-4′ carbon of the flavonoid skeleton [14,22,32,33]. Based on the experiments carried out here, the demethylation and 4″-O-methylglucosylation processes are the fastest when the methoxy group is located on the C-3′ carbon (Scheme 1)

### 2.4. 2′,5′-Dimethoxyflavone (**4**) Biotransformation

In the cultures of most of the tested strains the major product of biotransformation of 2′,5′-dimethoxyflavone (**4**) was the 4-*O*-methylglucosylated derivative (the same trend as was observed in case monomethoxy flavone biotransformations). As a result of up-scaled biotransformation of this substrate in the culture of *B. bassiana* KCh J1.5, three products were isolated: **8**, **15** and **16**. These compounds were also observed during biotransformation in other cultures (*B. bassiana* KCh J2.1 and BBT, *B. caledonica* KCh J3.2, and *I. fumosorosea* KCh J2). Compound **8**, with a retention time of 10.7 min (HPLC), was isolated in the highest yield (Table 5).

The ^1^H NMR spectrum shows signals from the 4-*O*-methylglucopyranosyl substituent. The HMBC spectrum shows the coupling of the signal from the H-1″ proton of the sugar unit with the signal from the C-5′ carbon of the flavone skeleton. The chemical shifts as well as the coupling of signals observed in the correlation spectra confirm that this product is 5′-*O*-β-D-(4″-*O*-methylglucopyranosyl)-2′-methoxyflavone (**8**). The next product is 2′-*O*-β-D-(4″-*O*-methylglucopyranosyl)-5′-methoxyflavone (**15**), the signal of which is observed on the HPLC chromatogram with a retention time of 11.6 min. NMR spectra indicate that it is also a product of demethylation and 4-*O*-methylglucosylation. The HMBC spectrum shows the coupling of the signal from the H-1″ proton of the sugar unit with the signal from the C-2′ carbon of the flavone skeleton, which indicates that this product is 2′-*O*-β-D-(4″-*O*-methylglucopyranosyl)-5′-methoxyflavone (**15**). The third product with the lowest yield in the culture of *B. bassiana* KCh J1.5 has also been identified as 4-*O*-methylglucopyranoside. The ^1^H NMR spectrum shows three singlets from the protons of the -OCH_3_ groups. One methoxy group is bonded to the C-4″ carbon of the sugar unit, the other two to the C-2′ and C-5′ carbons of the flavone skeleton. The B-ring protons of this product give only two singlets at 7.00 and 7.56 ppm. Based on the chemical shifts, multiplicities and couplings observed in the correlation spectra, they were assigned to the protons H-3′ and H-6′, respectively. The NMR spectra (Appendix A) made for this product indicate that it is 4′-*O*-β-D-(4″-*O*-methylglucopyranosyl)-2′,5′-dimethoxyflavone (**16**) (Figure 7).

Based on the NMR analyses of the obtained products, it can be seen that demethylation or hydroxylation and 4-*O*-methylglucosylation of 2′,5′-dimethoxyflavone (**4**) take place in the cultures of the strains tested. The highest concentration (over 70%) of 5′-*O*-β-D-(4″-*O*-methylglucopyranosyl)-2′-methoxyflavone (**8**) in the reaction mixture was observed in the culture of *I. fumosorosea* KCh J2. In the cultures of *B. bassiana* KCh J1.5, KCh J2.1, KCh J3.2 and *I. farinosa* KCh KW1.1 the concentration of 5′-*O*-β-D-(4″-*O*-methylglucopyranosyl)-2′-methoxyflavone (**8**) is almost twice as high as 2′-*O*-β-D-(4″-*O*-methylglucopyranosyl)-5′-methoxyflavone (**15**). However, in the cultures of *B. bassiana* KCh BBT and *B. caledonica* KCh J3.3 strains, the concentration of these products was comparable. In the culture of the *B. caledonica* KCh J3.4 strain, several products were observed, but they were formed at low concentrations that prevented their isolation.

Two 2′,5′-dimethoxyflavone (**4**) products were observed in the culture of the *B. bassiana* KCh J1 strain. The main product was identified as 5′-hydroxy-2′-methoxyflavone (**17**). Compared to the spectrum of the substrate, instead of two, only one singlet from the three protons (3.84 ppm) is visible in the ^1^H NMR spectrum of this product. At the same time, at 9.41 ppm, a singlet derived from the hydroxyl group is observed. The HMBC spectrum shows connections of this signal with signals from carbons C-4′, C-5′ and C-6′, which proves that the methoxyl group situated at the C-5′ carbon was demethylated.

The second product (**18**) is produced in a much smaller amount in the culture of the *B. bassiana* KCh J1 strain. The ^1^H NMR spectrum shows two signals from protons of -CH_3_ groups in positions 3.84 and 3.85 ppm, respectively, and a singlet derived from one proton, in the field characteristic for signals from Ar-OH protons. These data prove the preservation of both methoxy groups and the introduction of an additional hydroxyl group to the substrate molecule. The chemical shifts and multiplicities of the signals from the protons in the A ring are almost identical to those observed in the ^1^H NMR of the substrate spectrum. However, the B-ring protons of this product only give two singlets at 6.70 and 7.53 ppm. Based on the shapes and couplings observed in the correlation spectra, H-3′ and H-6′ were assigned, respectively. Additionally, in the HMBC spectrum, the signal from the proton of the hydroxyl group is coupled with the signals from carbons C-3′, C-4′ and C-5′. This information indicates that this compound is 4′-hydroxy-2′,5′-dimethoxyflavone (**18**) (Figure 8). In the culture of this strain, other products of hydroxylation were also formed in the seven-day biotransformation. However, it was not possible to isolate them and determine their structures.

### 2.5. 3′,4′,5′-Trimethoxyflavone (**5**) Biotransformation

The highest conversion (over 90% after the seventh day of the biotransformation process) of 3′,4′,5′-trimethoxyflavone was observed in the culture of *B. bassiana* KCh J1.5 and two strains of *B. caledonica* (KCh J3.3 and KCh J3.4). As a result of the scaled-up biotransformation in the culture of *B. bassiana* KCh J1.5, selected based on the results from HPLC (Table 6) and TLC, four products were isolated (**19–22**) (Figure 9). The main product is the effect of progressive demethylation and 4″-*O*-methylglucosylation. The ^1^H NMR spectrum shows all signals from the A-ring protons of the flavonoid skeleton with multiplicity and positions similar to the substrate spectrum (3′,4′,5′-trimethoxyflavone (**5**)). However, instead of two singlets, one for six protons, the other for three protons from the protons of the methoxy groups, three singlets from the protons of the -OCH_3_ groups are visible (two groups bound to the B ring of the flavone, one to the C-4″ carbon of the sugar unit). In addition, the signals from the H-2′and H-6′ protons in the substrate spectrum generated a singlet at 7.15 ppm due to the same chemical environment. The spectrum of this product shows two doublets at 7.41 ppm from H-6′ and 7.49 ppm from H-2′. Such differentiation of these signals proves that the substitution took place within the B ring. Based on the correlation spectra, it can be stated that the 4-*O*-methyl glucose substituent is bound to the C-3′ carbon (visible in the HMBC spectrum coupling of the signal from the H-1″ proton with the signal from carbon C-3′). The most common product of biotransformation of 3′,4′,5′-trimethoxyflavone (**5**) by *B. bassiana* KCh J1 was identified as 3′-*O*-β-D-(4″-*O*-methylglucopyranosyl)-4′,5′-dimethoxyflavone (**19**).

Furthermore, the second product is a 4-*O*-methylglucoside (**20**). However, the ^1^H NMR spectrum of **19** shows all signals from the flavonoid skeleton protons with multiplicity and positions similar to the 3′,4′,5′-trimethoxyflavone (**5**) spectrum. A singlet at 3.90 ppm derived from the six protons of methoxy groups (C-3′-OCH_3_ and C-5′-OCH_3_) is visible. The HMBC spectrum shows the coupling of the signal from the H-1″ proton with the signal from the C-4′ carbon, which proves that this product is 4′-*O*-β-D-(4″-*O*-methylglucopyranosyl)-3′,5′-dimethoxyflavone (**20**), which was identified in the cultures of most biocatalysts used. The main compound (**19**) was yielded in 80% of the reaction mixture in the culture of the *B. bassiana* KCh J1.5 strain and two *B. caledonica* strains (KCh J3.3 and KCh J3.4). In the cultures of these strains and the *B. bassiana* KCh J3.2 strain, the concentration of **19** was about five times higher than 4′-*O*-β-D-(4″-*O*-methylglucopyranosyl)-3′,5′-dimethoxyflavone (**20**). In the culture of *B. bassiana* KCh BBT, compounds **19** and **20** were formed in the ratio of 3:1, in the culture of *I. farinosa* KCh KW 1.1 in the ratio 10:1, and in the culture of *I*. *fumosorosea* KCh J2 in the ratio of 48:1.

In the culture of the *B. bassiana* KCh J1 strain, even after ten days of 3′,4′,5′-trimethoxyflavone (**5**) incubation, nearly 60% unreacted substrate was observed. The resulting products were formed in very low concentrations, which made them impossible to isolate. Instead, scaled-up biotransformation in the *B. bassiana* KCh J1.5 culture gave two additional products which are formed at very low concentrations in the cultures of most of the tested biocatalysts. The compound with a retention time of 10.3 min was identified as 6-*O*-β-D-(4″-*O*-methylglucopyranosyl)-3′,4′,5′-trimethoxyflavone (**21**).

Based on the correlation spectra recorded for compound **21**, it was found that the sugar unit is bound to carbon C-6. This compound is an effect of hydroxylation and glycosylation. Table 7 shows the positions of the signals visible in the ^13^C NMR spectrum of 6-*O*-β-D-(4″-*O*-methylglucopyranosyl)-flavone (**21**) and the flavone previously described, which were used as a calculation standard [14]. Additionally, the calculated chemical shifts caused by the introduction of the *O*-β-D-(4-*O*-methyl)-glucopyranosyl substituent on the C-6 carbon with respect to the position of analogous signals visible on the flavan spectrum are given. Table 7 shows the signal positions that are visible on the ^13^C NMR spectrum made for 3′,4′,5′-trimethoxyflavone (**5**) and the calculated signal positions for 6-*O*-β-D-(4″-*O*-methylglucopyranosyl)-3′,4′,5′-trimethoxyflavone (**21**). The ^13^C NMR data obtained for compound **21** are compatible with the calculated data (Table 7). Additionally, the structure of this product was confirmed by MS analysis (Appendix A).

Analogous calculations were made for product **22**. The chemical shifts of the carbon signals caused by the introduction of the hydroxyl group at C-6 were calculated. In this case, the positions of the signals on the ^13^C NMR spectrum of 6-hydroxy-flavone and flavone without any substituents, previously described [14], were compared and then the shift values were added to the values of the positions of the carbon signals from the 3′-*O*-β-D-(4″-*O*-methylglucopyranosyl)-4′,5′-dimethoxyflavone. Very high compatibility of the calculated and measured values of the signal positions in the ^13^C NMR spectrum for 3′-O-β-D-(4″-O-methylglucopyranosyl)-6-hydroxy-4′,5′-dimethoxyflavone (22) was obtained. The NMR and MS spectra confirm the correct determination of the product structure (Appendix A).

Flavonoid glycosylation in recent times has been increasingly described, which does not necessarily mean that we fully understood all the processes and enzymes responsible for these reactions. Nine strains of entomopathogenic filamentous fungi – *B. bassiana* KCh J1.5, J2.1, J3.2, J1, BBT, *B. caledonica* KCh J3.3, J3.4, *I. fumosorosea* KCh J2 and *I. farinosa* KCh KW 1.1 – used in these experiments were capable to transforming substrates containing a methoxy group(s) on ring B. Often the process of adding a sugar moiety to flavonoids is described as a one-step reaction, in which the end product is formed without the presence of intermediates [24,25,26,27,28,29,30]. Based on the structure of the products isolated here, it can be concluded that glycosylated derivatives of flavonoids are the result of cascade changes. During incubation in the studied strains, there takes place the processes of demethylation and/or hydroxylation and 4-*O*-methylglucosylation, in which several enzymes and coenzymes of the biocatalyst are involved. In addition, the position of the substituent within the B ring had a significant impact on the speed and number of products formed. The demethylation and 4-*O*-methylglucosylation processes are fastest when the methoxy group is placed on the C-3′ carbon and slowest when on the C-2′ carbon. However, the largest number of products were formed when the methoxy substituent was in the 2′- position, where the 4-*O*-methylglycose molecule was attached not only within the ring where it was present, but also outside it. The same was true in the case of the flavonoid containing 3 methoxy groups, where attachment of 4-*O-*methylglycose within another ring was also observed.

## 3. Materials and Methods

### 3.1. Substrates

The substrates 2-hydroxyacetophenone, 2-methoxybenzaldehyde, 3-methoxybenzaldehyde, 4-methoxybenzaldehyde, 2,5-dimethoxybenzaldehyde and 3,4,5-trimethoxybenzaldehyde were purchased from Sigma-Aldrich (St. Louis, MO, USA). Flavones used in biotransformations were synthesized from those substrates (the reactions are described below). The resulting chalcones were used as substrates for the flavone synthesis and their NMR spectral data are identical to those previously published [34,35,36].

### 3.2. Synthesis

All used substrates were synthesized in the laboratory by carrying out two kinds of reactions. Firstly five different methoxychalcones were synthesized in the Claisen-Schmidt reaction of 2-hydroxyacetophenone with suitable methoxybenzaldehyde in reaction, described earlier and shown in Scheme 2 [34,35,36,37,38]. The resulting flavones (**1**–**5**) were used as substrates for the biotransformation and their NMR spectral data are identical to those previously published [32,39].

After 2 h of reflux, the product of the Claisen-Schmidt reaction was transferred into an acid environment and filtered using a Buchner funnel. The obtained product (appropriate methoxychalcone) was confirmed by NMR analysis. All other methoxychalcones were synthesized analogously. Methoxyflavones were synthesized from methoxychalcones by reaction with iodine in DMSO with 2–3 h incubation (until the substrate has reacted completely) at 130 °C [40], as presented in the example below (Scheme 3 and Scheme 4).

All substrates for biotransformations were synthesized in the same way. The obtained compounds were confirmed by NMR (^1^H NMR, ^13^C NMR, COSY, HMBC and HSQC) analysis.

### 3.3. Microorganisms

The microorganisms *Beauveria bassiana* KCh J1.5, KCh J2.1, KCh J1, KCh J3.2 and KCh BBT, *B. caledonica* KCh J3.3 and KCh J3.4, *Isaria farinosa* KCh KW 1.1 and *I. fumosorosea* KCh J2. were obtained from the collection of the Department of Chemistry, Wrocław University of Environmental and Life Sciences (Wrocław, Poland). Isolation and identification procedures were described in our previous paper [21,22].

### 3.4. Screening Procedure

Erlenmeyer flasks (300 mL), each containing 100 mL of the sterile cultivation medium (3% glucose, 1% aminobac), were inoculated with a suspension of each entomopathogenic strain and then incubated for 3 days at 24 °C on a rotary shaker. After this time, 10 mg of a substrate was dissolved in 1 mL of dimethyl sulfoxide (DMSO) and added to the interior. Samples were collected on the 1st, 3rd, 7th and 10th day of the process. Then, all products were extracted using ethyl acetate, and extracts were dried using MgSO_4_, concentrated in vacuo and analyzed using TLC and HPLC methods.

#### Scale-up Biotransformation

For the scale-up process we used Erlenmeyer flasks (2000 mL), each containing 500 mL of the same cultivation medium (3% glucose, 1% aminobac), which were inoculated in the same way as described above. Three days after inoculation, 100 mg of a substrate was dissolved in 2 mL of DMSO and added to the interior. Samples were collected on the 14^th^ day of the process. Products were extracted three times using ethyl acetate and then analyzed using TLC, HPLC and NMR spectroscopy (^1^H NMR, ^13^C NMR, COSY, HMBC and HSQC) analysis.

### 3.5. Analysis

Initial tests were carried out using TLC plates (SiO_2_, DC Alufolien Kieselgel 60 F_254_ (0.2 mm thick), Merck, Darmstadt, Germany). The mobile phase contained a mixture of chloroform and methanol in 9:1 (*v*/*v*) relation. The plates were observed using a UV lamp (254 and 365 nm).

The scale-up biotransformation products were separated using 1000 µm preparative TLC silica gel plates (Anatech, Gehrden, Germany). The mobile phase contained a mixture of chloroform and methanol in a 9:1 (*v*/*v*) ratio. Separation products were scraped out and extracted twice using ethyl acetate.

#### 3.5.1. HPLC

A Waters 2690 instrument equipped with a Waters 996 photodiode array detector, using an ODS 2 column (4.6 × 250 mm, Waters, Milford, MA, USA) and a Guard-Pak Inserts µBondapak C18 pre-column was used to perform HPLC analyses. The mobile phase consisted of eluent A (80% acetonitrile in 4.5% acetic acid solution) and eluent B (4.5% acetic acid) with gradient elution: 0–7 min, 10% A/90% B; 7–10 min, 50% A/50% B; 10–13 min. 60% A/40% B; 15–20 min 80% A/20% B; 20–30 min, 90% A/10% B; 30–40 min, 100% A. The flow rate was 1.0 mL/min, injection volume was 10 µL, detection wavelength 280 nm.

##### UHPLC

A Thermo Scientific Dionex Ultimate 3000 UHPLC+ instrument (Thermo Scientific, Waltham, MA, USA) with a photodiode array detector (detection in wavelength: 210–450 nm) with a C-18 analytical column ZORBAX Eclipse XDB (5 µm, 4.6 × 250 mm, Agilent, Santa Clara, CA, USA) was used to performed UHPLC analyses. Chromatographic separation was achieved using an isocratic elution of 50% A (0.05% formic acid water solution) and 50% B (methanol containing 0.05% of formic acid) for 2 min, then a linear gradient of B from 50% to 95% for 10 min and isocratic elution of 95% B for 5 min. The flow rate was 1.0 mL/min.

#### 3.5.2. NMR Spectroscopy

The NMR analysis was performed with a DRX 600 MHz Bruker spectrometer (Bruker, Billerica, MA, USA) with an UltraShield Plus magnet and measured in DMSO-d_6_ or Acetone-d_6_.

##### 2′-*O*-β-D-(4″-*O*-Methylglucopyranosyl)-Flavone (**6**)

^1^H NMR (600 MHz) (DMSO) δ (ppm): 3.07 (dd, 1H, *J* = 9.5, 9.1 Hz, H-4″), 3.29-3.35 (m, 1H, H-2″), 3.41-3.54 (m, 3H, H-3″, H-5″and one of H-6″), 3.46 (s, 3H, C-4″-OC*H*_3_), 3.64 (ddd, 1H, *J* = 11.6, 4.9, 1.6 Hz, one of H-6″), 4.72 (dd, 1H, *J* = 6.2, 5.0 Hz, C-6″-O*H*), 5.16 (d, 1H, *J* = 7.8 Hz, H-1″), 5.27 (d, 1H, *J* = 5.8 Hz, 3″-O*H*), 5.39 (d, 1H, *J* = 5.5 Hz, C-2″-O*H*), 7.08 (s, 1H, H-3), 7.22 (ddd, 1H, *J* = 7.3, 6.9, 1.0 Hz, H-5′), 7.35 (dd, 1H, *J* = 8.6, 0.8 Hz, H-3′), 7.50 (ddd, 1H, *J* = 8.1, 7.1, 1.1 Hz, H-6), 7.50 (ddd, 1H, *J* = 8.1, 7.1, 1.1 Hz, H-4′), 7.74 (dd, 1H, *J* = 8.5, 0.7 Hz, H-8), 7.83 (ddd, 1H, *J* = 8.7, 7.1, 1.7 Hz, H-7), 7.93 (dd, 1H, *J* = 7.9, 1.7 Hz, H-6′), 8.06 (ddd, 1H, *J* = 7.9, 1.6, 0.4 Hz, H-5). ^13^C NMR (151 MHz, DMSO) δ = 59.74 (C-4″-O*C*H_3_), 60.15 (C-6″), 73.50 (C-2″), 75.70 (C-5″), 76.55 (C-3″), 78.89 (C-4″), 99.73 (C-1″), 112.17 (C-3), 115.37 (C-3′), 118.58 (C-8), 120.72 (C-1′), 122.00 (C-5′), 123.20 (C-4a), 124.75 (C-5), 125.37 (C-6), 129.26 (C-6′), 132.70 (C-4′), 134.22 (C-7), 155.30 (C-2′), 156.01 (C-8a), 160.37 (C-2), 177.27 (C-4).

##### 8-*O*-β-D-(4″-*O*-Methylglucopyranosyl)-2′-Methoxyflavone (**7**)

^1^H NMR (600 MHz) (Acetone-d_6_) δ (ppm): 3.27 (dd, 1H, *J* = 9.7, 8.9 Hz, H-4″), 3.53 (ddd, 1H, *J* = 9.7, 4.7, 2.1 Hz, H-5″), 3.61 (dd, 1H, *J* = 12.5, 4.4 Hz, H-2″), 3.64-3.74 (m, 2H, H-3″ and one of H-6″), 3.85 (ddd, 1H, *J* = 10.9, 4.8, 2.0 Hz, one of H-6″), 4.03 (s, 3H, C-4″-OC*H*_3_), 5.18 (d, 1H, *J* = 7.5 Hz, H-1″), 7.10 (s, 1H, H-3), 7.16 (ddd, 1H, *J* = 7.9, 7.4, 1.1 Hz, H-5′), 7.25 (dd, 1H, *J* = 8.4, 0.9 Hz, H-3′), 7.35 (t, 1H, *J* = 8.0 Hz, H-6), 7.56 (ddd, 1H, *J* = 8.4, 7.4, 1.8 Hz, H-4′), 7.61 (dd, 1H, *J* = 8.1, 1.5 Hz, H-7), 7.74 (dd, 1H, *J* = 8.0, 1.5 Hz, H-5), 8.22 (dd, 1H, *J* = 7.9, 1.7 Hz, H-6′). ^13^C NMR (151 MHz, Acetone-d6) δ = 56.36 (C-4″-O*C*H_3_), 60.58 (C-4″-O*C*H_3_), 61.99 (C-6″), 75.02 (C-2″), 77.21 (C-5″), 78.13 (C-3″), 80.01 (C-4″), 102.24 (C-1″), 112.68 (C-3), 113.06 (C-3′), 118.59 (C-5), 120.84 (C-7), 121.26 (C-1′), 121.83 (C-5′), 125.43 (C-6), 125.76 (C-4a), 130.59 (C-6′), 133.51 (C-4′), 147.70 (C-8a), 148.02 (C-8), 159.23 (C-2′), 160.88 (C-2), 178.08 (C-4).

##### 5′-*O*-β-D-(4″-*O*-Methylglucopyranosyl)-2′-Methoxyflavone (**8**)

^1^H NMR (600 MHz) (DMSO) δ (ppm): 3.02 (t, 1H, *J* = 9.4 Hz, H-4″), 3.26 (td, 1H, *J* = 9.0, 4.0 Hz, H-2″), 3.40-3.45 (m, 2H, H-3″, H-5″), 3.46 (s, 3H, C-4″-OC*H*_3_), 3.53 (ddd, 1H, *J* = 11.6, 6.2, 5.8 Hz, one of H-6″), 3.67 (ddd, 1H, *J* = 11.4, 4.7, 1.4 Hz, one of H-6″), 3.90 (s, 3H, C-2″-OC*H*_3_), 4.77 (t, 1H, *J* = 5.7 Hz, C-6″-O*H*), 4.87 (d, 1H, *J* = 7.8 Hz, H-1″), 5.29 (d, 1H, *J* = 5.4 Hz, C-3″-O*H*), 5.43 (d, 1H, *J* = 4.5 Hz, C-2″-O*H*), 6.99 (s, 1H, H-3), 7.20 (d, 1H, *J* = 9.2 Hz, H-3′), 7.27 (dd, 1H, *J* = 9.1, 3.1 Hz, H-4′), 7.50 (ddd, 1H, *J* = 8.0, 7.0, 1.0 Hz, H-6), 7.63 (dd, 1H, *J* = 3.0, Hz, H-6′), 7.76 (dd, 1H, *J* = 7.9, 0.6 Hz, H-8), 7.83 (ddd, 1H, *J* = 8.6, 7.0, 1.6 Hz, H-7), 8.05 (dd, 1H, *J* = 7.9, 1.6 Hz, H-5). ^13^C NMR (151 MHz, DMSO) δ = 56.39 (C-2′-O*C*H_3_), 59.71 (C-4″-O*C*H_3_), 60.39 (C-6″), 73.48 (C-2″), 75.73 (C-5″), 76.28 (C-3″), 79.20 (C-4″), 101.11 (C-1″), 111.71 (C-3), 113.68 (C-3′), 116.93 (C-6′), 118.61 (C-8), 120.14 (C-1′), 120.88 (C-4′), 123.10 (C-4a), 124.71 (C-5), 125.45 (C-6), 134.33 (C-7), 151.17 (C-5′), 152.88 (C-2′), 155.91 (C-8a), 160.05 (C-2), 177.18 (C-4).

##### 3-*O*-β-D-(4″-*O*-Methylglucopyranosyl)-2′-Methoxyflavone (**9**)

^1^H NMR (600 MHz) (DMSO) δ (ppm): 3.07 (t, 1H, *J* = 9.3 Hz, H-4″), 3.27-3.37 (m, 2H, H-2″ and H-5″), 3.40 (s, 3H, C-4″-OC*H*_3_), 3.41-3.55 (m, 2H, H-6″), 3.79 (s, 3H, C-3-OC*H*_3_), 4.45 (t, 1H, *J* = 5.3 Hz, C-6″-O*H*), 5.14 (d, 1H, *J* = 5.5 Hz, 3″-O*H*), 5.27 (d, 1H, *J* = 7.8 Hz, H-1″), 5.31 (d, 1H, *J* = 4.7 Hz, C-2″-O*H*), 7.06 (td, 1H, *J* = 7.4, 0.8 Hz, H-5′), 7.16 (d, 1H, *J* = 8.3 Hz, H-3′), 7.50-7.55 (m, 2H, H-4′ and H-6), 7.66 (dd, 1H, *J* = 8.4, 0.4 Hz, H-8), 7.73 (dd, 1H, *J* = 7.6, 1.7 Hz, H-6′), 7.82 (ddd, 1H, *J* = 8.6, 7.1, 1.7 Hz, H-7), 8.14 (dd, 1H, *J* = 8.0, 1.4 Hz, H-5).

##### 2′-Hydroxyflavone (**10**)

^1^H NMR (600 MHz) (Acetone-d_6_) δ (ppm): 7.07 (ddd, 1H, *J* = 8.0, 7.3, 1.1 Hz, H-5′), 7.13 (dd, 1H, *J* = 8.2, 1.0 Hz, H-3′), 7.19 (s, 1H, H-3), 7.42 (ddd, 1H, *J* = 8.2, 7.3, 1.7 Hz, H-4′), 7.48 (ddd, 1H, *J* = 8.1, 7.1, 1.1 Hz, H-6), 7.70 (ddd, 1H, *J* = 8.4, 1.1, 0.5 Hz, H-8), 7.80 (dd, 1H, *J* = 8.8, 6.9, 1.6 Hz, H-7), 7.99 (dd, 1H, *J* = 7.9, 1.7 Hz, H-5), 10.03 (s, C-2′-O*H*). ^13^C NMR (151 MHz, Acetone-d_6_) δ = 112.54 (C-3), 117.93 (C-3′), 119.18 (C-8), 119.58 (C-1′), 120.91 (C-5′), 124.71 (C-4a), 125.85 (C-6), 125.89 (C-5), 129.73 (C-6′), 133.25 (C-4′), 134.62 (C-7), 157.16 (C-2′), 157.32 (C-8a), 161.76 (C-2), 178.25 (C-4).

##### 3′-Hydroxyflavone (**11**)

^1^H NMR (600 MHz) (DMSO) δ (ppm): 6.94 (s, 1H, H-3), 7.01 (ddd, 1H, *J* = 8.1, 2.5, 0.9 Hz, H-4′), 7.38 (t, 1H, *J* = 7.9 Hz, H-5′), 7.44 (t, 1H, *J* = 1.9 Hz, H-2′), 7.51 (ddd, 1H, *J* = 8.0, 6.9, 1.1 Hz, H-6), 7.53 (ddd, 1H, *J* = 7.8, 1.8, 1.0 Hz, H-6′), 7.77 (ddd, 1H, *J* = 8.4, 1.2, 0.5 Hz, H-8), 7.86 (ddd, 1H, *J* = 8.5, 6.9, 1.6 Hz, H-7), 8.06 (ddd, 1H, *J* = 7.9, 1.7, 0.5 Hz, H-5), 9.90 (s, C-3′-O*H*). ^13^C NMR (151 MHz, DMSO) δ = 106.93 (C-3), 112.85 (C-2′), 117.23 (C-6′), 118.53 (C-8), 118.88 (C-4′), 123.36 (C-4a), 124.82 (C-5), 125.57 (C-6), 130.29 (C-5′), 132.45 (C-1′), 134.38 (C-7), 155.68 (C-8a), 157.92 (C-3′) 162.73 (C-2), 177.10 (C-4).

##### 3′-*O*-β-D-(4″-*O*-Methylglucopyranosyl)-Flavone (**12**)

^1^H NMR (600 MHz) (DMSO) δ (ppm): 3.05 (t, 1H, *J* = 9.2 Hz, H-4″), 3.25-3.34 (m, 1H, H-2″), 3.42-3.57 (m, 3H, H-3″, H-5″and one of H-6″), 3.47 (s, 3H, C-4″-OC*H*_3_), 3.68 (dd, 1H, *J* = 10.0, 4.9 Hz, one of H-6″), 4.80 (t, 1H, *J* = 5.2 Hz, C-6″-O*H*), 5.07 (d, 1H, *J* = 7.8 Hz, H-1″), 5.31 (d, 1H, *J* = 5.5 Hz, C-3″-O*H*), 5.46 (d, 1H, *J* = 5.2 Hz, C-2″-O*H*), 7.07 (s, 1H, H-3), 7.26 (dd, 1H, *J* = 8.2, 2.4 Hz, H-4′), 7.47-7.55 (m, 2H, H-5′ and H-6), 7.73 (d, 1H, *J* = 2.0 Hz, H-2′), 7.76 (d, 1H, *J* = 7.8 Hz, H-8), 7.79-7.87 (m, 2H, H-6′ and H-7), 8.06 (d, 1H, *J* = 7.9 Hz, H-5). ^13^C NMR (151 MHz, DMSO) δ = 59.76 (C-4″-O*C*H_3_), 60.35 (C-6″), 73.48 (C-2″), 75.75 (C-5″), 76.34 (C-3″), 79.19 (C-4″), 100.02 (C-1″), 107.24 (C-3), 113.84 (C-2′), 118.69 (C-6′), 119.79 (C-4′), 120.00 (C-8), 123.36 (C-4a), 124.81 (C-5), 125.63 (C-6), 130.31 (C-5′), 132.52 (C-1′), 134.43 (C-7), 155.72 (C-8a), 157.81 (C-3′), 162.25 (C-2), 177.23 (C-4).

##### 4′-Hydroxyflavone (**13**)

^1^H NMR (600 MHz) (DMSO) δ (ppm): 6.88 (s, 1H, H-3), 6.92-6.96 (m, 2H, H-3′, H-5′), 7.48 (ddd, 1H, *J* = 8.0, 7.0, 1.2 Hz, 1H, H-6), 7.75 (ddd, 1H, *J* = 8.4, 1.2, 0.4 Hz, H-8), 7.81 (ddd, 1H, *J* = 8.5, 6.9, 1.7 Hz, H-7), 7.95-7.99 (m, 2H, H-2′, H-6′), 8.03 (dd, 1H, *J* = 7.9, 1.7, 0.4 Hz, H-5), 10.32 (s, C-4′-O*H*). ^13^C NMR (151 MHz, DMSO) δ = 104.83 (C-3), 115.97 (C-3′ and C-5′), 118.39 (C-8), 121.60 (C-1′), 123.34 (C-4a), 124.75 (C-5), 125.35 (C-6), 128.39 (C-2′ and C-6′), 134.06 (C-7), 155.61 (C-8a), 161.00 (C-4′), 163.09 (C-2), 176.91 (C-4).

##### 4′-*O*-β-D- (4″-*O*-Methylglucopyranosyl)-Flavone (**14**)

^1^H NMR (600 MHz) (DMSO) δ (ppm): 3.06 (t, 1H, *J* = 9.3 Hz, H-4″), 3.29 (ddd, 1H, *J* = 9.1, 7.8, 5.2 Hz, H-2″), 3.42-3.49 (m, 2H, H-3″, H-5″), 3.47 (s, 3H, C-4″-OC*H*_3_), 3.52 (ddd, 1H, *J* = 11.5, 6.4, 5.1 Hz, one of H-6″), 3.65 (ddd, *J* = 11.4, 5.0, 1.5 Hz, one of H-6″), 4.74 (dd, 1H, *J* = 6.2, 5.1 Hz, C-6″-O*H*), 5.07 (d, 1H, *J* = 7.8 Hz, H-1″), 5.30 (d, 1H, *J* = 5.5 Hz, C-3″-O*H*), 5.48 (d, 1H, *J* = 5.2 Hz, C-2″-O*H*), 6.99 (s, 1H, H-3), 7.18-7.22 (m, 2H, H-3′, H-5′), 7.50 (ddd, 1H, *J* = 8.1, 6.8, 1.3 Hz, H-6), 7.78 (ddd, 1H, *J* = 8.5, 1.3, 0.4 Hz, H-8), 7.83 (ddd, 1H, *J* = 8.5, 6.8, 1.7 Hz, H-7), 8.04 (ddd, 1H, *J* = 7.9, 1.6, 0.5 Hz, H-5), 8.06-8.10 (m, 2H, H-2′, H-6′). ^13^C NMR (151 MHz, DMSO) δ = 59.74 (C-4″-O*C*H_3_), 60.18 (C-6″), 73.37 (C-2″), 75.69 (C-5″), 76.25 (C-3″), 78.94 (C-4″), 99.50 (C-1″), 105.76 (C-3), 116.54 (C-3′ and C-5′), 118.50 (C-8), 123.33 (C-4a), 124.41 (C-5), 124.78 (C-6), 125.46 (C-1′), 128.11 (C-2′ and C-6′), 134.20 (C-7), 155.65 (C-8a), 160.05 (C-4′), 162.46 (C-2), 177.02 (C-4).

##### 2′-*O*-β-D-(4″-*O*-Methylglucopyranosyl)-5′-Methoxyflavone (**15**)

^1^H NMR (600 MHz) (DMSO) δ (ppm): 3.05 (t, 1H, *J* = 9.4 Hz, H-4″), 3.25-3.29 (m, 1H, H-2″), 3.40-3.45 (m, 2H, H-3″, H-5″), 3.45 (s, 3H, C-4″-OC*H*_3_), 3.51-3.55 (m, 1H, one of H-6″), 3.63 (ddd, 1H, *J* = 12.1, 4.8, 1.7 Hz, one of H-6″), 3.81 (s, 3H, C-6″-OC*H*_3_), 4.69 (d, 1H, *J* = 5.5 Hz, C-6″-O*H*), 5.03 (d, 1H, *J* = 7.8 Hz, H-1″), 5.24 (d, 1H, *J* = 5.8 Hz, C-3″-O*H*), 5.34 (d, *J* = 5.6 Hz, C-2″-O*H*), 7.13 (dd, 1H, *J* = 9.2, 3.2 Hz, H-4′), 7.30 (d, 1H, *J* = 9.2 Hz, H-3′), 7.44 (d, 1H, *J* = 3.1 Hz, H-6′), 7.50 (ddd, 1H, *J* = 8.1, 7.0, 1.1 Hz, H-6), 7.77 (dd, 1H, *J* = 8.3, 0.7 Hz, H-8), 7.83 (ddd, 1H, *J* = 8.6, 7.1, 1.8 Hz, H-7), 8.05 (dd, 1H, *J* = 7.9, 1.8 Hz, H-5). ^13^C NMR (151 MHz, DMSO) δ = 55.73 (C-2′-O*C*H_3_), 59.72 (C-4″-O*C*H_3_), 60.21 (C-6″), 73.55 (C-2″), 75.70 (C-5″), 76.56 (C-3″), 78.96 (C-4″), 100.56 (C-1″), 112.36 (C-3), 113.87 (C-6′), 117.13 (C-3′), 117.99 (C-4′), 118.71 (C-8), 121.58 (C-1′), 123.20 (C-4a), 124.71 (C-5), 125.37 (C-6), 134.18 (C-7), 149.34 (C-5′), 153.94 (C-2′), 155.99 (C-8a), 160.14 (C-2), 177.27 (C-4).

##### 4′-*O*-β-D-(4″-*O*-Methylglucopyranosyl)-2′,5′-Dimethoxyflavone (**16**)

^1^H NMR (600 MHz) (DMSO) δ (ppm): 3.00 (t, 1H, *J* = 9.2 Hz, H-4″), 3.32-3.37 (m, 1H, H-2″), 3.39-3.43 (m, 1H, H-5″), 3.46 (s, 3H, C-4″-OC*H*_3_), 3.47-3.54 (m, 2H, H-3″ and one of H-6″), 3.64-3.69 (m, 1H, one of H-6″), 3.84 (s, 3H, C-5″-OC*H*_3_), 3.90 (s, 3H, C-2″-OC*H*_3_), 4.79 (t, 1H, *J* = 5.1 Hz, C-6″-O*H*), 5.14 (d, 1H, *J* = 7.9 Hz, H-1″), 5.32 (d, 1H, *J* = 5.6 Hz, C-3″-O*H*), 5.49 (d, 1H, *J* = 5.6 Hz, C-2″-O*H*), 6.98 (s, 1H, H-3), 7.00 (s, 1H, H-3′), 7.48 (ddd, 1H, *J* = 8.0, 7.1, 1.3 Hz, H-6), 7.56 (s, 1H, H-6′), 7.79 (dd, 1H, *J* = 8.4, 1.3 Hz, H-8), 7.81 (ddd, 1H, *J* = 8.5, 7.1, 1.5 Hz, H-7), 8.03 (dd, 1H, *J* = 7.9, 1.5 Hz, H-5). ^13^C NMR (151 MHz, DMSO) δ = 56.41 (C-2′-O*C*H_3_), 56.69 (C-6′-O*C*H_3_), 59.75 (C-4″-O*C*H_3_), 60.39 (C-6″), 73.27 (C-2″), 75.96 (C-5″), 76.67 (C-3″), 79.36 (C-4″), 99.48 (C-1″), 101.40 (C-3′), 110.56 (C-3), 111.82 (C-1′), 113.06 (C-6′), 119.60 (C-8), 123.11 (C-4a), 124.66 (C-5), 125.29 (C-6), 134.33 (C-7), 142.81 (C-4′), 150.23 (C-5′), 153.23 (C-2′), 155.78 (C-8a), 160.31 (C-2), 177.12 (C-4).

##### 5′-Hydroxy-2′-Methoxyflavone (**17**)

^1^H NMR (600 MHz) (DMSO) δ (ppm): 3.84 (s, 3H, C-2′-OC*H*_3_), 6.94 (s, 1H, H-3), 6.97 (dd, 1H, *J* = 8.9, 3.0 Hz, H-4′), 7.09 (d, 1H, *J* = 9.0 Hz, H-3′), 7.32 (d, 1H,*J* = 3.0 Hz, H-6′), 7.50 (ddd, *J* = 8.0, 7.1, 0.8 Hz, H-6), 7.71 (d, 1H, *J* = 8.2 Hz, H-8), 7.83 (ddd, 1H, *J* = 8.6, 7.2, 1.7 Hz, H-7), 8.05 (dd, 1H, *J* = 7.9, 1.4 Hz, H-5), 9.41 (s, C-5′-O*H*). ^13^C NMR (151 MHz, DMSO) δ = 56.33 (C-2′-O*C*H_3_), 111.57 (C-3), 114.07 (C-3′), 115.01 (C-6′), 118.43 (C-8), 119.36 (C-4′), 120.29 (C-1′), 123.13 (C-4a), 124.77 (C-5), 125.42 (C-6), 134.33 (C-7), 150.80 (C-2′), 151.12 (C-5′), 155.89 (C-8a), 160.53 (C-2), 177.16 (C-4).

##### 4′-Hydroxy-2′,5′-Dimethoxyflavone (**18**)

^1^H NMR (600 MHz) (DMSO) δ (ppm): 3.84 (s, 3H, C-5′-OC*H*_3_), 3.85 (s, 3H, C-2′-OC*H*_3_), 6.70 (s, 1H, H-3′), 6.95 (s, 1H, H-3), 7.47 (ddd, 1H, *J* = 8.1, 6.4, 1.9 Hz, H-6), 7.53 (s, 1H, H-6′), 7.78 (ddd, 1H, *J* = 8.6, 7.2, 1.7 Hz, H-7), 7.81 (d, 1H, *J* = 8.2 Hz, H-8), 8.02 (dd, 1H, *J* = 7.9, 1.5 Hz, H-5), 10.07 (s, C-4′-O*H*). ^13^C NMR (151 MHz, DMSO) δ = 56.17 (C-5′-O*C*H_3_), 56.64 (C-2′-O*C*H_3_), 101.11 (C-3′), 109.36 (C-1′), 109.90 (C-3), 112.87 (C-6′), 118.51 (C-8), 123.11 (C-4a), 124.63 (C-5), 125.16 (C-6), 133.89 (C-7), 141.87 (C-4′), 151.41 (C-5′), 153.86 (C-2′), 155.74 (C-8a), 160.67 (C-2), 177.08 (C-4).

##### 3′-*O*-β-D-(4″-*O*-Methylglucopyranosyl)-4′,5′-Dimethoxyflavone (**19**)

3.04 (t, 1H, *J* = 9.1 Hz, H-4″), 3.31-3.36 (m, 1H, H-2″), 3.48 (s, 3H, C-4″-OC*H*_3_), 3.46-3.58 (m, 3H, H-3″, H-5″ and one of H-6″), 3.70 (dd, 1H, *J* = 9.9, 5.0 Hz, one of H-6″), 3.81 (s, 3H, C-4′-OC*H*_3_), 3.91 (s, 3H, C-5′-OC*H*_3_), 4.84 (t, 1H, *J* = 5.5 Hz, C-6″-O*H*), 5.07 (d, 1H, *J* = 7.8 Hz, H-1″), 5.32 (d, 1H, *J* = 5.6 Hz, C-3″-O*H*), 5.46 (d, 1H, *J* = 5.7 Hz, C-2″-O*H*), 7.11 (s, 1H, H-3), 7.41 (d, 1H, *J* = 2.0 Hz, H-6′), 7.49 (ddd, 1H, *J* = 8.1, 5.3, 2.9 Hz, H-6), 7.54 (d, 1H, *J* = 2.0 Hz, H-2′), 7.81-7.84 (m, 2H, H-7 and H-8), 8.04 (ddd, 1H, *J* = 8.0, 1.5, 0.6 Hz, H-5). ^13^C NMR (151 MHz, DMSO) is presented in Table 7.

##### 4′-*O*-β-D-(4″-*O*-Methylglucopyranosyl)-3′,5′-Dimethoxyflavone (**20**)

3.03 (t, 1H, *J* = 9.1 Hz, H-4″), 3.31-3.36 (m, 1H, H-2″), 3.44 (s, 3H, C-4″-OC*H*_3_), 3.46-3.58 (m, 3H, H-3″, H-5″ and one of H-6″), 3.69 (dd, 1H, *J* = 9.9, 5.0 Hz, one of H-6″), 3.90 (s, 6H, C-3′-OC*H*_3_ and C-5′-OC*H*_3_), 4.52 (dd, 1H, *J* = 5.9, 4.9 Hz, C-6″-O*H*), 5.14 (d, 1H, *J* = 7.6 Hz, H-1″), 5.19 (d, 1H, *J* = 5.5 Hz, C-3″-O*H*), 5.23 (d, 1H, *J* = 4.9 Hz, C-2″-O*H*), 7.14 (s, 1H, H-3), 7.39 (s, 2H, H-2′ and H-6′), 7.49 (m, 1H, H-6), 7.81-7.84 (m, 2H, H-7 and H-8), 8.04 (ddd, 1H, *J* = 8.0, 1.5, 0.6 Hz, H-5). ^13^C NMR (151 MHz, DMSO) δ = 56.76 (C-3′-O*C*H_3_ and C-5′-O*C*H_3_), 59.66 (C-4″-O*C*H_3_), 60.34 (C-6″), 74.40 (C-2″), 76.08 (C-5″), 76.42 (C-3″), 79.18 (C-4″), 101.90 (C-1″), 104.89 (C-2′, and C-6′), 106.78 (C-3), 118.73 (C-8), 123.33 (C-4a), 124.77 (C-5), 125.53 (C-6), 126.27 (C-1′), 134.21 (C-7), 137.35 (C-4′), 152.93 (C-3′ and C-5′), 155.71 (C-8a), 162.44 (C-2), 177.22 (C-4).

##### 6-*O*-β-D-(4″-*O*-Methylglucopyranosyl)-3′,4′,5′-Trimethoxyflavone (**21**)

^1^H NMR (600 MHz) (DMSO) δ (ppm): 3.08 (t, 1H, *J* = 9.3 Hz, H-4″), 3.28 (ddd, 1H, *J* = 9.1, 7.3, 4.7 Hz, H-2″), 3.36-3.39 (m, 1H, H-5″), 3.41-3.46 (m, 2H, H-3″ and one of H-6″), 3.47 (s, 3H, C-4″-OC*H*_3_), 3.62-3.67 (m, 1H, one of H-6″), 3.75 (s, 3H, C-4′-OC*H*_3_), 3.91 (s, 6H, C-3′-OC*H*_3_ and C-5′-OC*H*_3_), 4.74 (dd, 1H, *J* = 6.4, 5.0 Hz, C-6″-O*H*), 5.01 (d, 1H, *J* = 7.8 Hz, H-1″), 5.28 (d, 1H, *J* = 5.5 Hz, C-3″-O*H*), 5.46 (d, 1H, *J* = 5.2 Hz, C-2″-O*H*), 7.14 (s, 1H, H-3), 7.39 (s, 2H, H-2′ and H-6′), 7.51 (dd, 1H, *J* = 9.1, 3.1 Hz, H-7), 7.58 (d, 1H, *J* = 3.1 Hz, H-5), 7.83 (d, 1H, *J* = 9.2 Hz, H-8). ^13^C NMR (151 MHz, DMSO) is presented in Table 7.

##### 3′-*O*-β-D-(4″-*O*-Methylglucopyranosyl)-6-Hydroxy-4′,5′-Dimethoxyflavone (**22**)

^1^H NMR (600 MHz) (DMSO) δ (ppm): 3.03 (t, 1H, *J* = 9.1 Hz, H-4″), 3.30-3.35 (m, 1H, H-2″), 3.47 (s, 3H, C-4″-OC*H*_3_), 3.43-3.57 (m, 3H, H-3″, H-5″ and one of H-6″), 3.68-372 (m, 1H, one of H-6″), 3.80 (s, 3H, C-4′-OC*H*_3_), 3.90 (s, 3H, C-5′-OC*H*_3_), 4.81 (t, 1H, *J* = 5.4 Hz, C-6″-O*H*), 5.05 (d, 1H, *J* = 7.9 Hz, H-1″), 5.29 (d, 1H, *J* = 5.7 Hz, C-3″-O*H*), 5.43 (d, 1H, *J* = 5.7 Hz, C-2″-O*H*), 7.02 (s, 1H, H-3), 7.25 (dd, 1H, *J* = 9.0, 3.0 Hz, H-6), 7.31 (d, 1H, *J* = 3.0 Hz, H-8), 7.38 (d, 1H, *J* = 2.0 Hz, H-6′), 7.50 (d, 1H, *J* = 2.0 Hz, H-2′), 7.68 (d, 1H, *J* = 9.0 Hz, H-5), 10.02 (s, 1H, -O*H*). ^13^C NMR (151 MHz, DMSO) is presented in Table 7.

#### 3.5.3. LC-MS

Molecular formulas of products were confirmed by LC-MS 8045 SHIMADZU analysis. The mobile phase was a mixture of 0.1% aqueous formic acid v/v (A) and acetonitrile (B). The program was as follows: 80% B and 20% A in 5 min. The flow rate was 0.3 mL/min and the injection volume was 2 µl. The column (Kinetex 2.6 µm C18 100 Å, 100 mm x 3 mm, Phenomenex, Torrance, CA, USA) was operated at 30 °C. The major operating parameters were as follows: nebulizing gas flow: 3 L/min, heating gas flow: 10 L/min, interface temperature: 300°C, drying gas flow: 10 L/min, data acquisition range m/z 100–1000 Da; ionization mode—positive. Data were collected with LabSolutions (Shimadzu, Kyoto, Japan) software.

## 4. Conclusions

To sum up, obtaining 18 novel flavonoids via whole-cell fungal biotransformations was described, 13 of them being 4-*O*-methylglucosides and 5 hydroxy flavones. To the best of our knowledge, these flavone glucosides have never been reported before.

Our previous studies have shown that the ability of 4-*O*-methylglycosyl attachment to the hydroxyl group in the flavonoid molecule is unique for entomopathogenic filamentous fungi. In this publication, we have demonstrated the capacity of these strains to produce 4-*O*-methylglucoside flavonoids, the formation of which is preceded by the effective *O*-demethylation and/or hydroxylation of the methoxyflavones obtained by chemical synthesis.

The performed biotransformations resulted in products that may indicate that glycosylation is not a one-step process, but is likely the result of an enzyme cascade as intermediates also appear. Obtaining these products is, therefore, possible due to the use of whole cells cultures of biocatalysts containing many interacting enzymes.

Some of the entomopathogenic fungi used (e.g., *B. bassiana* KCh J1.5 or *I. fumosorosea* KCh J2) were able to introduce a sugar group into the flavone ring by hydroxylation and then 4-*O*-methylglycosylation. This sequence resulted in four different glycosylated products in a one-pot reaction. The demethylation and 4-*O*-methylglucosylation processes are fastest when the methoxy group is on the C-3′ carbon and slowest when the methoxy substituent is on the C-2′ carbon. However, placing the substituent at the C-2′ position results in the highest number of end products. We have obtained 13 new glycosylated flavones (4-*O*-methylglucopyranosides) and five hydroxy derivatives. Each of the obtained products was determined by means of HPLC, LC-MS and NMR (Appendix A).

Additionally, research on the properties of flavonoids, including their glycoside derivatives, is still ongoing, and efficient methods for obtaining such compounds are sought continuously. The methods presented in this publication allow significant amounts of glycoside derivatives to be obtained in an efficient and relatively cheap way while following the principles of “green chemistry”.

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
