# Peer review of "Biotransformation of Methoxyflavones by Selected Entomopathogenic Filamentous Fungi"

_ijms, 2020, doi:10.3390/ijms21176121_

Round 1

Reviewer 1 Report

This paper describes the biotransformation of 5 different methoxy flavones by 8 different strains/species of entomopathogenic fungus. For the most part the results indicated that the flavone was initially demethylated and then glycosylated with a 4-O-methylglucopranosyl group. In most instances the major product was the replacement of the methyl group by the 4-O-methylglucopyranosyl group. Minor di (and poly) glocsylated products were identified when the biotransformation introduced a new oxygen.

The paper is well written in English. There is perhaps to much emphasis put on the structure determination on some compounds when they only have the loss of a methyl group and addition of a 4-0-methylglocopyranosyl group. (The structures are quite simple, and spectral appearances obvious.) 

In most journals, there is a requirement to establish the chemical formula of a reported compound. This is usually done via an elemental analysis or through mass spectrometry using a high resolution instrument. A quadrupole instrument does not give the chemical formula. (It can be inferred through proper analysis of the isotopic patter, but the instrument only gives unit resolution.)

Author Response

We thank the Reviewer for his inspirational evaluation of our manuscript.

According to the suggestion, we corrected the conclusions section. In our opinion, the current form of this chapter is now more precise and better summarizes the results described in the manuscript.

We agree with the Reviewer that performing HRMS analysis or elemental analysis is essential in the structural analysis of chemical compounds. However, our research unit does not have such equipment, and commissioning such analyzes exceeds our financial capacity. For these reasons, in our research, we focused on determining the catalytic abilities of selected strains of filamentous fungi. The use of HPLC with a UV detector with a high probability allows for a preliminary determination of the structure of flavonoid compounds. Additionally, the biotransformation products are unambiguously characterized by NMR methods, and their mass has been confirmed by MS analysis. The use of this methodology allows characterizing the transformations of the tested flavonoid compounds by the biocatalysts used, and at the same time, reducing to the absolute minimum doubts about the characteristics of the described relationships.

Reviewer 2 Report

This is an excellent paper on synthesis of five methoxyflavones and their biotransformation by whole-cell of five strains entomopathogenic fungi. As a result, eighteen novel flavone glucosides have been reported for the first time. Biotransformation of natural compounds is a very powerful method for generation and screening of new drug candidates with potential application in pharmacy. Even that the topic is a bit aside from the journal scope, I think it would be of interesting for a wide range of researchers working on chemistry of natural compounds, pharmacy, analytical chemistry and biochemistry. In general, the paper is written in high scientific language, the results and conclusions are clearly presented and supported by huge raw data sets, provided as supplementary materials. 

Author Response

We thank the Reviewer for giving our manuscript a very favourable opinion. Such an assessment is very inspiring for us (as for every person conducting research), it reassures us that the research we conduct is important and adds enthusiasm for planning and carrying out further experiments.